# Clicker Training as an Applied Refinement Measure in Chickens

**DOI:** 10.3390/ani13243836

**Published:** 2023-12-13

**Authors:** Gordon Mählis, Anne Kleine, Dörte Lüschow, Alexander Bartel, Mechthild Wiegard, Christa Thoene-Reineke

**Affiliations:** 1Institute of Animal Welfare, Animal Behavior and Laboratory Animal Science, School of Veterinary Medicine, Freie Universität Berlin, Königsweg 67, 14163 Berlin, Germany; mechthild.wiegard@fu-berlin.de (M.W.); thoene-reineke.christa@fu-berlin.de (C.T.-R.); 2Division for Poultry, Farm Animal Clinic, School of Veterinary Medicine, Freie Universität Berlin, Königsweg 63, 14163 Berlin, Germany; anne.kleine@fu-berlin.de (A.K.); doerte.lueschow@fu-berlin.de (D.L.); 3Institute of Veterinary Epidemiology and Biostatistics, School of Veterinary Medicine, Freie Universität Berlin, Königsweg 67, 14163 Berlin, Germany

**Keywords:** hen, animal welfare, training of animals, animal behavior, animal husbandry, immobilization, stress, clicker training, positive reinforcement training, corticosterone, metabolic profiling

## Abstract

**Simple Summary:**

As part of a larger study, corticosterone levels in plasma and saliva as well as the assessment of behavior and fecal output in a New Area Test were used in chickens used for experimental purposes to investigate the possibility of mitigating the stress sensation, which is easily induced by handling and sampling by means of prior three-week clicker training. It was shown that this training is a suitable measure of cognitive enrichment in the sense of the 3R principle and that the first stress impulse at the beginning of the stress exposure can be attenuated. However, three-week clicker training is neither sufficient for a sustainable reduction in stress perception in prolonged stress situations nor for the sustainable strengthening of self-confidence and resilience. Further studies with longer training periods, more stressful situations, and additional parameters of stress assessment are warranted.

**Abstract:**

When using chickens in animal studies, the handling of these animals for sample collection or general examinations is considered stressful due to their prey nature. For the study presented here, plasma and salivary corticosterone as well as New Area Test behavior and fecal output were used to evaluate whether it is possible to influence this stress perception using a three-week clicker training program. The results indicate that clicker training seems to be a suitable refinement measure in the sense of cognitive enrichment for the husbandry of this species. However, since it was also shown that three-week training was not sufficient to sustainably reduce the stress perception with regard to prolonged stressor exposure, and since it was also evident that manipulations such as routine blood sampling are perceived as less stressful than assumed, further studies with prolonged training intervals and situations with higher stressor potential are warranted. Also, further parameters for training assessment must be considered. For the general use of training as a supportive measure in animal experiments, its proportionality must be considered, particularly considering the expected stress and adequate training time.

## 1. Introduction

For years, numerous studies have been devoted to the three principles of *replacement*, *reduction*, and *refinement* with regard to the ethical optimization of the use of animals in scientific experiments [1,2,3,4,5,6,7]. Conceived by the British zoologist William Russel and his colleague, the microbiologist Rex Burch, and published in their book “The Principles of Humane Experimental Technique” in 1959, this 3R principle is now an integral part of European and German legislation [8,9,10].

In addition to the constant attempt to replace animal experiments with animal-free methods and the continuous endeavor to reduce the number of animals used in indispensable animal experiments to the minimum required for validity, the third parameter, refinement, optimizes the standards in all areas of scientific animal use in order to avert or at least minimize potential pain, suffering, or permanent harm [1,5,11,12]. With regard to refinement, the constant optimization of animal husbandry and improvements in animal experimental methods have taken place for many years [12,13,14]. Despite animals’ frequent use in experiments, to this day, only a few findings in this regard have slipped into the focus of the discussion [15]. In recent years, particularly clicker training has proven to be a promising method of refinement [3,16,17]. According to Thorndike and Skinner, this type of training is based on the process of “operant conditioning”, which stems from the behavioral sciences and that is still popular today, not only in training pets and companion animals [18,19,20], but is also increasingly being used in the context of medical training in zoos and laboratory animal facilities [3,16,17,21,22,23,24,25]. As part of a scientific research project with chickens in 2018, the function of the trainer was replaced by the use of a semi-automatic training machine and, thus, further standardized as part of a scientific application [26]. During the execution of a certain behavior desired by the trainer in the context of clicker training, the animal is encouraged to perform this behavior more often by means of a click sound and subsequent reward, usually provided as a favored food item. This also allows the animal to learn to perform this in response to a signal from the trainer. The click sound is considered a conditioned reinforcer for the animal using classical conditioning prior to the actual training, by which the animal then reliably anticipates the reward, i.e., the unconditional stimulus, in the actual clicker training phase in which the click sound becomes a conditioned stimulus or secondary reinforcer [27]. In operant conditioning, any previously neutral stimulus, e.g., a light stimulus or the acoustic signal of a whistle or a spoken word [28,29,30], can become a secondary reinforcer. However, in clicker training, it is usually the sound of the eponymous clicker device.

With regard to the more than 7.9 million laboratory animals used for experimental purposes in the EU and Norway in 2020, birds still accounted for more than 500,000 animals (6.43%) [31]. In the UK, birds accounted for 8.91% of the experimental animals used in 2022, with over 134,000 out of a total of over 1.5 million animals [32]. Of the laboratory animals used in the EU and Norway, most of them—almost 1.5 million—are used in Germany [31]. Regarding the numbers of laboratory animals used here and published by the German Federal Institute for Risk Assessment in 2021, the most commonly used birds here are chickens, quails, and zebra finches [33]. While there is a high demand for chickens for microbiological and immunological studies, vaccine development, or basic research [33,34], this species is also often used for educational purposes, e.g., in university veterinary teaching. The capture, handling, and sampling procedures required in experimental projects, in and of themselves, represent a significant stress factor for this prey species [35,36,37]. Since birds have been shown to be in no way inferior to mammals on a psychological level and in terms of social intelligence [38], the testing of clicker training intended to reduce stress in chickens used for experimental purposes aims at taking into account refinement and, thus, the welfare of these species in laboratory animal science.

The influence of clicker training on the animals’ perception can be tested individually on an animal basis using various laboratory diagnostic and ethological parameters. A parameter readily used to assess stress perception in birds is the stress hormone corticosterone, which is predominant in these animals [39,40,41,42,43,44]. Although Ericsson et al. [40] recognized that the corticosterone-mediated stress response of wild Bankiva chickens has been attenuated over the course of their domestication, all forms of immobilization remain among the most significant stressors in the handling of chickens and can cause a corticosterone increase after only two minutes [40,44]. Knowles and Broom [37] obtained higher corticosterone levels in chickens that were crated prior to blood sampling than from animals sampled without delay and directly from their cages. The data of the basal concentration considered physiological for blood (serum or plasma) corticosterone in chicken ranges from 0.14 to 20 ng/mL [45,46,47,48,49,50,51,52,53,54,55,56].

For mammalian cortisol/corticosterone, studies show a relatively strong positive correlation between the values obtained from plasma versus saliva samples [57,58,59]. Although Vincent and Michell [58] found only 4 to 10% of the plasma corticosterone concentration in saliva, Teruhisa et al. [59] found 70%, while Greenwood and Shutt [60] found equal concentrations of free cortisol in blood and saliva. Kirschbaum and Hellhammer [61] suggest that this correlation may be due to steroids very easily diffusing through the cell membranes of the salivary gland cells due to their low molecular weight and good lipid solubility.

Since, in addition to the wide range of concentrations, no uniform procedure has been established with regard to corticosterone extraction and the various assays available work differently, the measured corticosterone concentrations should always be interpreted as relative values—e.g., in a comparison of two groups—and not as absolute values [62]. SCANES [63] has identified the need for a recommendable supplement to the serological parameter of corticosterone in relation to the assessment of stress perception in chickens used in behavioral studies.

A behavioral test often used to assess stress or anxiety in animals that are prey animals by nature is the open field test. This behavioral test, developed by the American psychologist Calvin Springer Hall, was originally designed to assess the behavior of laboratory rodents and is still widely used today due to its ease of implementation [64,65]. In this test, the animal to be examined—originally mostly a rat or mouse—is placed in a free, unfamiliar space (an open field or new area) and its handling of this situation, which it would naturally avoid to prevent attacks by predators, is assessed over a certain period of time. Depending on the size of the animal or the existing local conditions, simple boxes or free spaces (New Area Test) can also be used in which (depending on the scientific question) the different behaviors exhibited by the animal, such as the distance covered in a certain time, can be recorded [64].

In addition to behavioral parameters, the assessment of the feces produced in the New Area Test can also aid in assessing the chickens’ perception of stress. After the intestinal motility abates during the first and sympathetic-mediated segment of a stress situation, the release of corticotropin-releasing hormone (CRH) in the course of a prolonged and non-compensable stress can stimulate the intestinal mast cells. Mast cells contain many different messenger substances—first and foremost, histamine—and are of particular importance in type I allergies as well as in the defense against different pathogens and toxins [66,67,68,69]. In addition to the immunological mast cell degranulation through the IgE produced by plasma cells typical for type I allergy, the CRH released in the context of non-compensable stress reactions may mediate a similar effect. The mast cells stimulated by CRH release a large number of paracrine substances, which stimulate the enteric nervous system and control blood flow, peristalsis, absorption, and secretion [70]. This stimulus is intended to trigger the expulsion of intestinal contents in stress situations and, thus, to accelerate the elimination of pathogens. The link between a stressful situation and visceral hypersensitivity [71], usually accompanied by diarrhea and described as hyperacute for birds [72], is known as the brain–gut axis [73] and may be used as a solid assessment criterion of the animals’ respective stress perception.

Using the combination of serological and ethological testing, the aim of this study was to investigate whether chickens can be trained using clicker training so as to experience less stress in stressful situations such as during blood and saliva sampling or handling during a general examination. To assess the animals’ acute stress sensation, it was investigated whether clicker training leads to a reduction in the rapid corticosterone level increase in blood and saliva occurring in acute stress situations in what can be considered an indicator of lower stress perception. A retrospective assessment of the stress sensation during handling and sampling was performed by evaluating the feces deposited by the chickens in the New Area Test with regard to the number of fecal droppings and their consistency. Furthermore, it was investigated whether the training can lead to a higher level of resilience, which was assessed by the axial body shaking as “rearrangement behavior” in the New Area Test. As a second behavioral parameter, the locomotion rate of each animal, i.e., its effort to reunite with the group, was assessed and, thus, the potentially positive influence of previous clicker training on the animals’ self-confidence could be identified.

## 2. Material and Methods

### 2.1. Ethics Statement

The experiment was approved by the local authorities (State Office of Health and Social Affairs Berlin, approval ID G 0328/18) and conducted in accordance with the German Animal Welfare Act and the German Laboratory Animal Protection Ordinance.

### 2.2. Animals

The animals used in this study were conventionally farm-raised laying hens of the hybrid line Lohmann Selected Leghorn, which were kept under the same conditions, and fed and administered the same basic vaccination scheme. At the age of 18–19 weeks, they were imported into our husbandry and housed in an aviary. Upon import, each animal was marked individually with a colored ring, weighed, and submitted to general examination, during which the typical state of health parameters for the chickens were assessed. These included the inspection of all animals with regard to their nutritional and care status, potential injuries, inflammation, the presence of infections, or an infestation with ectoparasites. After confirming the health of the animals, they were included in the study. After being housed in the aviaries, the animals were given a two-week acclimatization phase in which they could become used to the new situation. After these two weeks, the first samples were taken and the training of the training group animals began. Three consecutive experimental runs were carried out, each with seven training and seven control animals. All seven animals from each group were housed together in the same aviary. Both groups were housed in adjacent aviaries and had visual contact with each other. In their aviaries, the animals were provided with laying nests, perches on two levels, wood shavings as bedding, a pecking stone, and other materials of environmental enrichment (pecking toys, fodder beets hanging free on a chain). The animals had access to an ad libitum supply of water and conventional laying hen feed.

### 2.3. Training

The training schedule of every experimental run covered three weeks. The training of the individual animals was suspended as soon as they showed signs of lacking motivation. However, an interruption of one hour was generally sufficient to regain the training readiness of the animals. The entire training of all three training groups was carried out by the same person. The animals were trained so that they were prepared for all types of handling (e.g., palpation of body parts) or immobilization (for taking blood or saliva samples) that would occur in the later general examination representing the stressful situation. The group training took place daily and consisted of 6–8 training sessions of 20–30 min each (depending on the severity of the respective training step and the motivation of the animals). The single training that followed in the second training phase consisted of 4–5 training sessions per day and usually lasted 10–15 min per animal. This also depended on the severity of the respective training step and the animal’s motivation.

### 2.4. Training Procedure

#### 2.4.1. Reward Food and Clicker–Reward Combination

Due to its taste and worm-like appearance, considered attractive by chickens, grated cheese is a highly suitable reward. In order to avoid the rapid saturation of the animals by consuming larger amounts of cheese, it was mixed with different grains, i.e., pigeon feed. In order to have one hand free for animal handling, a clicker–reward combination was constructed using an empty plastic cup (diameter 9 cm, height 4 cm), adhesive tape (duct tape), a tablespoon, and the clicker gadget. The plastic cup was taped to the convex (back) side of the spoon bowl and the clicker gadget to its handle without obstructing the “clicker plate” (see attached photos).

#### 2.4.2. Influence of Chicken Breed and Individual Character on Training

Chickens are not as accustomed to the immediate proximity and touch of humans as, for example, companion animal species such as dogs or cats. In addition, the Lohmann Selected Leghorn hybrids used in this study are known to react more sensitively to handling stress than other frequently used hybrid lines, such as the Lohmann Brown [74]. Therefore, the individual training steps were adapted to this circumstance and sometimes deviated slightly between different individuals depending on whether they were more proactive or reactive learners. Since the chickens were trained in a group during the first training phase, the individual training steps and days trained may have overlapped due to the different training speeds of the individual animals.

#### 2.4.3. Group Training

Day 1: The aviary door was opened by the trainer, but the aviary was not entered. The clicker was triggered and food was scattered in the aviary immediately.

Day 2: The aviary was entered by the trainer, the clicker was triggered, and food was scattered in the aviary immediately.

Day 3: The aviary was entered and the trainer took a seat on a stool to further reduce the distance. The clicker was triggered and food was scattered in the aviary immediately.

Day 4: Sitting on the stool, the trainer no longer scattered the food when the click sound was produced, but it was offered to the birds directly by hand or with the clicker–reward combination and by the immediate triggering of the clicker.

Day 5: Initially, the training was the same as on day 4. As soon as all of the birds gathered around the trainer, first touches were exerted by placing an empty hand in the immediate vicinity of the chickens, which then regularly touched the chickens casually. When this happened, the clicker sounded immediately and the respective casually touched chicken was given an extra reward with food.

Days 6 and 7: The chickens were increasingly actively touched in the breast, flank, and back areas.

Days 7 and 8: A second stool was placed next to the existing stool so that the animals could fly onto it, thus further reducing the distance between them and the trainer. Oftentimes, some of the animals moved from the second stool or, over time, jumped directly from the floor to the trainer’s lap. Each new step made by the individual animals was rewarded with an immediate click.

Days 9–12: After five animals had jumped onto the trainer’s lap, a table was positioned in front of the aviary door and used as seating for the trainer. Again, the animals jumped onto the trainer’s lap, this time covering a greater distance. As soon as all of the animals had flown onto the trainer’s lap, the trainer positioned himself behind the table. Again, the animals then flew towards the trainer, automatically directly onto the table. At the end of the training phase, the chickens flew onto the table, regardless of where the trainer positioned it and himself in the room.

#### 2.4.4. Single Training

Days 13–16: A more in-depth training of animal handling and weighing on a scale located on the table was then carried out for the individual animals. Initially, the trainer touched the wings or flanks of the animal being trained. Over the course of the three days, this was then increasingly supplemented by short but forced touches under the wings or lifting of the wings, which both became longer and longer over time. Also, touches of the head and cloacal area were included. Lifting of the birds was trained by reaching under the breast and lifting the animals slowly. While the acceptance of the handling and lifting was being trained, the animal also regularly entered the scale on its own initiative. Whenever this happened, it was positively rewarded and thus also trained.

Days 17–23: The chickens learned to move from the table to the trainer’s hand independently. The acceptance of the trainer’s turning movements and sampling including all individual steps, such as fixing the legs, turning the animal sideways, positioning the wings, disinfecting the wings, handling colored blood sample tubes, and touching the beak, and the presence of and touching by a second person was then trained and consolidated until the end of the training period.

### 2.5. Sample Collection, Sample Preparation, and Description of the General Examination as the Induced Stress Situation

A total of three blood and saliva samples were collected from each animal. The average sampling time per animal was 2.5 min, during which first the blood sample and then the saliva sample were taken. The first sample collection occurred after the 14-day acclimation period, i.e., prior the training period. To minimize the sample collection time in the first sampling in which all individuals of one experimental run were sampled on the same day, two teams of two were used for sampling. One team sampled the training group and the other team sampled the control group. The second and third sampling took place after the three-week training period on the day the animals were exposed to the 10 min stress situation of a general examination, i.e., directly prior to and after this handling on the examination table. This was performed on seven consecutive days, each at 2 p.m., for one randomly selected animal from the training group and its ring color counterpart from the control group. The general examination of the animals on the table as the induced stress situation included the inspection and palpation of the abdominal and cloacal area, inspection and palpation of the feet for pododermatitis, inspection and palpation of the head appendages, weighing of the animal, and inspection and palpation of the undersides of the wings and the uropygial gland. Sampling 2 and 3 were always carried out by the same two persons for all animals. The training group animals were led to the sampling according to the clicker program they had learned, while the control animals had to be caught out of their aviary. The procedure of sample collection and preparation followed the same scheme in all three experimental runs. The blood samples were taken via the ulnar vein after the disinfection of the puncture site. The sample tube was then inverted 10 times and labeled. For blood collection and anticoagulation, 4 mL Li-heparin tubes were used and filled with 3 mL of blood for each sample. The tubes were then centrifuged for 10 min at room temperature and 2000 g and the plasma obtained was pipetted into 1.5 mL centrifuge tubes at 150 µL each. The aliquots were then frozen at −80 °C until further analysis. The saliva samples were obtained by swabbing the sublingual and inner cheek area of the respective animal’s beak for one minute. If present, macroscopic contamination of the cotton swab was removed. The swab was inserted into a 1.5 mL centrifuge tube using a clamp constructed according to NEMETH et al. [75]. The swab stick was removed and the tubes containing the swab itself were centrifuged at 10,900× *g* for 10 min in order for the saliva to settle at the bottom of the tubes. After centrifugation, the clamp, together with the swab, was removed and the samples were frozen at −80 °C until analysis.

### 2.6. Corticosterone Measurement

#### 2.6.1. Verification of Corticosterone ELISA

Corticosterone levels were determined using a Corticosterone ELISA Kit (Neogen^®^, Ayr, Scottland, UK) and the associated analysis protocol. The Microplate Absorbance Reader SUNRISE (TECAN, Crailsheim, Germany) and the Magellan evaluation software (SW MAGELLAN V7.2 STD5PC XP VISTA WIN7/8, TECAN, Grödig, Austria) were used to retrieve and evaluate the data. Optical density was determined at a wavelength of 450 nm using the reference wavelength of 620 nm. The ELISA used in this study was verified by determining intra- and inter-assay precision prior to sample analysis. The intra-assay was performed by measuring three plasma samples in three duplicates each. For the inter-assay, two single duplicate assays prepared separately were analyzed and the randomly selected second of the three intra-assay duplicate values was added as the third value for each sample. Verification of the ELISA with respect to saliva samples was not performed due to the very small sample volume. In the subsequent analysis, all plasma or saliva samples from one animal, as well as equal numbers of samples from training and control group animals, were analyzed on the same ELISA plate.

#### 2.6.2. Extraction of Plasma Corticosterone

For plasma corticosterone extraction, the NEOGEN protocol was followed, modified only for the resuspension after the evaporation step according to Freeman and Newman [76] by replacing 5% of the resuspension buffer with ethanol. According to Freeman and Newman, this method aims to facilitate the dissolution of the hormone in the aqueous buffer, consequently reducing the variance between the duplicates in the subsequent analysis.

#### 2.6.3. Dilution of Saliva Sample

Saliva corticosterone did not need to be extracted prior to analysis according to the NEOGEN ELISA protocol. However, since the sample volumes were often insufficient for duplicate determination, enough material was extracted from each saliva sample to achieve the necessary sample volume through a fivefold dilution.

### 2.7. New Area Test—Behavior and Fecal Matter

To assess stress perception during the handling previously experienced and to evaluate the self-confidence of the trained versus untrained animals during social isolation, all animals were placed in the corridor in front of the stable room containing the aviaries for a period of 5 min after the last sampling and filmed. Later, these video sequences were coded and randomly analyzed by two institute employees. Behavioral evaluation was performed with respect to axial body shaking as well as with respect to the distance traveled in animal lengths during the observation period (Table 1). The mean value of the observations made by the two institute employees was then calculated for each parameter.

In addition to the behavioral parameters, the fecal droppings of the animals of both groups were also assessed regarding number and consistency directly after the completion of the respective tests (Table 2).

### 2.8. Statistical Evaluation

The statistical evaluation was carried out with SPSS 25. First, a test for normal distribution was performed analytically using the Shapiro–Wilk test and graphically using the Q-Q plots. The values of corticosterone concentrations in the plasma and saliva samples were logarithmized before testing, as already established [77]. If testing with or without logarithmization showed a normal distribution, further analysis using an independent t-test followed. If the tests showed a lack of normal distribution, the Mann–Whitney U test was used for further analysis. Following this approach, the values of plasma corticosterone were analyzed using the independent t-test. For the salivary corticosterone values, the tests did not indicate a normal distribution, possibly due to corticosterone concentration levels below the detection limit in either one (1× control group and 2× training group) or even both (1× control group and 1× training group) parts of the respective duplicate set of the saliva samples analyzed. In these cases, the minimum concentration of 0.001 ng/mL was assumed. Four further samples (1: sample 1 of a control group animal; 2–4: samples 1–3 of a training group animal)) were excluded from the evaluation due to a technical error during sample analysis. Based on the results regarding normal distribution, salivary corticosterone analysis was performed using the Mann–Whitney U test. The normal distributed values of locomotion in the New Area Test were also analyzed with the independent *t*-test, while body shaking and fecal parameters were analyzed using the Mann–Whitney U test. For both the independent *t*-test and the Mann–Whitney U test, the significance level was set at *p* < 0.05.

## 3. Results

### 3.1. Clicker Training

#### 3.1.1. Group Training

During the group training of the chickens, an interplay between different aspects, such as differences in the individual learning speed, the individual animal characters, or the respective hierarchy level of the individual animal, ensured that the animals varied in their training progress. At these moments, it became necessary to allow high-ranking, fast-learning animals into the area in front of the aviaries. If they were kept busy there with provided food, the remaining animals were able to catch up on their training backlog. At other times, such as flying onto the second stool, the trainer’s lap, or the treatment table, it was helpful to leave individual animals with a high willingness to train in the group as “role models” and to motivate the other group members to imitate their training successes. If individual animals nevertheless hesitated to complete certain training steps, the training goal could be achieved by incorporating intermediate steps. For example, some animals were more reluctant to cover the long distance from the stable floor to the treatment table by flying up. These animals were then helped to fly onto the treatment table by using a stool as an intermediate step.

#### 3.1.2. Single Training

During the individual training of the single steps for handling and sample collection, parameters such as the respective animals’ body tension or the lack of “rescue attempts”, which could be shown by grasping the trainer’s hand/arm with the claws or flying off the treatment table, were particularly important criteria for recognizing a currently trained single step as being successfully completed. If an animal no longer tried to grasp the trainer’s hand or the cup containing the reward food and instead let its feet hang down without any tension, the training step was considered successfully completed. In addition to the group hierarchy, which could especially play a role during group training (pushing of lower-ranking animals out of the training environment by higher-ranking animals), the differences in the respective animal characters became particularly clear during individual animal training, which sometimes made it necessary to adapt the training program.

#### 3.1.3. Character Types, Description of the Character, and Their Frequency in This Study

Type 1—courageous and active (frequency: often (9 of 21 animals))

Continuous training was possible after becoming used to the clicker and the trainer in the first days. There was quick learning of actions, such as “flying onto the table” or “climbing onto the hand”. There was a high degree of initiative on the part of the animals. The animals were often briefly irritated when touched or manipulated by the trainer. The training motivation of the animals was high.

Type 2—courageous and accepting (frequency: often (8 of 21 animals))

After becoming used to the clicker and the trainer in the first days, continuous training was possible. There was quick learning success with regard to the acceptance of touch or manipulation by the trainer. Less initiative was shown by the animals with regard to actions, but after motivation by the trainer, there was also willingness to perform actions. The training motivation of the animals was high.

Type 3—fearful and active (frequency: rare (3 of 21 animals))

Even though the animal’s motivation for training was quite high, the training was often discontinuous and characterized by the animal’s constant ambition to participate in training and frequent, abrupt training interruptions or setbacks after the animal escaped from the training situation. Training could continue after breaks, but the overall training effort was high.

Type 4—fearful (frequency: very rare (1 of 21 animals))

The animal’s motivation for training was low. If training was possible at all, it was very discontinuous and patchy. The animal usually escaped from the situation during the first training steps (e.g., flying onto a table). Generally, a large distance from the trainer was maintained.

### 3.2. Corticosterone ELISA

#### 3.2.1. Verification of Corticosterone ELISA for Plasma

The intra-assay of the corticosterone ELISA showed a variation coefficient (VC) of 15.3% geometric mean. The determined variation coefficient for the inter-assay was 17.5% geometric mean.

#### 3.2.2. Corticosterone Concentrations in Plasma and Saliva

A comparison of the two groups based on their plasma corticosterone levels (Table 3, Figure 1) showed a significant group difference in sample 1, which was obtained before the start of training. In samples 2 and 3, a group difference that described lower plasma corticosterone values for the training group animals than for the control group animals was found only as a non-significant and—at the latest in plasma sample 3—minimal tendency. For the salivary corticosterone values (Table 4, Figure 1), on the other hand, the group difference recognizable in plasma sample 1 was only a tendency, while in saliva sample 2, it was significant with regard to lower salivary corticosterone values in the training group. The Pearson correlation coefficient [78] was used to assess this group difference, attributing a medium strength to the effect evident here. In the course of the subsequent handling, for saliva sample 3, both groups equalized so that here, too, only a tendency and not a significant group difference was shown with regard to lower salivary corticosterone values in the training group animals.

### 3.3. New Area Test Behavior

Concerning the amount of axial body shaking (Table 5), the training and control groups differed significantly with a mean effect size of 0.4 [78]. With regard to locomotion in the New Area Test (Table 5), the group differences were not significant.

Control group:-One animal: disturbance by crowing rooster in a neighboring stable.

Training group:-Two animals: disturbance by crowing rooster in a neighboring stable.-Two animals: temporarily in unavoidable blind spot of camera.

### 3.4. New Area Test—Defecation

No significant group difference in terms of the *p*-value was found with regard to the number of droppings deposited, but differences were found with regard to fecal consistency (both Table 6). One animal of the training group did not defecate at all and was therefore not considered in the evaluation of defecation consistency. The significant group difference in fecal consistency showed a high effect size.

## 4. Discussion

Training according to the protocol described in this study has positive effects on the stress response and behavior of chickens in a laboratory animal science setting and, thus, may represent applied refinement as cognitive enrichment in the sense of the 3Rs. However, whether the sustainable strengthening of stress resistance and self-confidence is possible must be verified using further investigation parameters and also tested for longer training periods and situations with different stress potential.

### 4.1. Verification of Corticosterone ELISA

The verification of the corticosterone ELISA on the basis of the coefficients of variation shows that corticosterone values are relatively susceptible to fluctuations. This high susceptibility to fluctuations was also suggested in the study conducted by Ännais Carbajal [79], who used the same ELISA and reported an intra-assay and inter-assay VC of about 10 and 15%, respectively, and noted how difficult it is to lower it into the single-digit range. Therefore, in this study, all samples of the same sample type and of one animal and an equal number of samples from training and control animals were analyzed on the same ELISA plate.

In addition, it is known that many factors, such as too-cold or too-warm ambient temperatures [80,81,82,83,84], incorrect lighting [85], stocking density or ammonium concentration [86,87,88,89], laying period, vaccinations, feeding regime, or parasites [35,49,90,91,92,93] may influence concentrations of corticosterone. While care was taken in animal housing, daily housing, and health management to mitigate the effect of other stressors as much as possible, certain parameters, such as hierarchy within groups, may have contributed to a higher variance of certain individual hormone levels [35,49,80,81,82,83,84,85,86,87,88,89,90,91,92,93].

Since this known potential for fluctuation makes it difficult to assess the stress perception alone, the subsequent New Area Test was performed based on the recommendation of SCANES [63], who suggests that additional behavioral tests should be carried out to substantiate corticosterone measurements.

### 4.2. Stress Measurement Using Plasma and Salivary Corticosterone

The first stress stimulus exerted on the animals in this study on the day of the second and third sampling was their individualization from their respective groups and being placed in the appropriate positions for the second plasma and saliva sampling, i.e., fixed in the wing (for blood) or beak (for saliva) area, and then sampled (sample 2). The means of individualization differed between the groups, i.e., the animals voluntarily leaving their aviary and flying onto the treatment table in the training group versus being caught out of the aviary in the respective control group. As expected, fixation was associated with more resistance in the control than in the training animals, who had been conditioned to this effect through clicker training.

The time between the opening of the respective aviary door and the completion of the first blood and saliva sample collection was approximately 2.5 min. The blood sample was taken first and lasted about 1.5 min. This was followed by a one-minute saliva sample collection. CHLOUPEK et al. [44] recommended blood sampling from the chicken within the first 2 min after the capture due to the possible increase in corticosterone levels. DERELI FIDAN et al. [40] also found an increase in plasma corticosterone levels in chickens restrained for 120 s.

The positive effect of the three-week clicker training becomes apparent in the salivary corticosterone levels determined for the second sampling, which are significantly lower in the trained group than in the untrained control group. Although this group difference still tends to be recognizable in saliva sample 3, both groups have equalized towards this point and the significance is lost. In addition, this group difference is also apparent in plasma samples 2 and 3, but is insignificant in both cases, and with regard to sample 3, there is only very minimal tendency. Besides this, and in considering the results of the verification with some caution, the results suggest that three weeks of clicker training may mitigate the effects of the initial stress impulse or reduce the resulting rapid surge of corticosterone. However, with regard to the values of the saliva or plasma samples taken after handling, the three-week training program is not sufficient to maintain a significantly reduced stress sensation over the course of a longer-lasting stress situation, as presented in this study using a ten-minute handling/examination procedure followed by sample collection. It must also be considered that the situation of handling and sampling in this study was apparently not perceived by all animals as being as stressful as is often generally assumed. This becomes evident by comparing the results of this study with the values for basal plasma corticosterone concentration in other studies, which range between 5 ng/µL and >20 ng/mL [36,42,45,46,47]. In order to evaluate the positive effect of clicker training with regard to a refinement measure in the handling of chickens in laboratory animal science, this study should be followed by further studies in which larger stress stimuli are set or a longer training period than three weeks should precede them. In addition to considering additional parameters to evaluate a longer training period or larger stressful stimuli, corticosterone measurement should also be performed again to see whether the results presented here can be confirmed or even strengthened.

With regard to the collection of the saliva sample, it should be considered that this intervention is perceived as a greater burden than wing vein puncture due to the necessity of extensive beak and head fixation. The animals in the control group showed relatively stronger defense movements such as wriggling or head retraction during saliva versus blood sampling. In the training group, the training toward the acceptance of saliva sampling was also more time consuming than toward blood sampling. Hence, although saliva sampling in larger animals such as ruminants or dogs may be classified as not or less stressful due to its non-invasiveness [58,94,95,96,97], this assessment may not apply for species such as chickens or other small animals and must be considered in a differentiated manner.

The results that have to be considered restrictively in this study are those obtained by the first plasma sampling, which indicated significant differences between the training and control group animals. However, these samples should not have exhibited any differences between the two groups at this time point, i.e., prior to the training of the experimental group animals. Up to this point, both groups had been kept under the exact same conditions during the two-week acclimation period. This discernible group difference could be explained by the very rapid response of the chickens’ plasma corticosterone levels to capture and immobilization stress [98]. As a prey animal with a correspondingly strong escape reflex, prior capture, sampling per se, as well as the accompanying immobilization can cause a rapid, stress-induced increase in corticosterone levels in this species [35,37]. For the first sampling, all 14 animals in each experimental run were sampled on the same day and consecutively captured from their aviaries for this purpose. In order to avoid a consecutive increase in the plasma corticosterone values from one chicken to the next due to perceiving the sampling stress inflicted on the conspecifics, two sampling teams, one allocated to the training and the other to the control group, were implemented in order to reduce the total sampling time as far as possible. Each team consisted of one person catching and fixing the animals and a second person for the sampling procedure itself. The implementation of two sampling teams reduced the total sampling time by half, but a team-related (and, hence, group-related) difference in terms of stress perception and, thus, plasma corticosterone levels cannot be excluded. The only other possible measure of shortening the actual sampling time would have been crating the animals prior to sampling and sampling them from this box in rapid succession. However, in this regard, KNOWLES and BROOM [37] found that the corticosterone levels of chickens crated prior to blood sampling were higher than those of animals sampled directly from their cages.

### 4.3. New Area Test

#### 4.3.1. Shaking as a Reordering Behavior

The shaking behavior observed in the New Area Test was not interpreted as a comfort behavior similar to grooming or scratching, but rather as a reaction to an individual disturbance in the context of the preceding capture, immobilization, and handling procedure. Maintaining or restoring individual body integrity is described by SCHMIDT [99] as an active process and was consequently interpreted as a “behavior of reorganization” in the context of the New Area Test conducted here and has also been described, for example, in agility dogs after stressful competition situations [100]. As chickens are similar to mammals in terms of their psychological level [38], it was expected that the chickens in our test would also “shake themselves back” to their natural integrity after the previous stressful situation. This behavior can also be observed in chickens after other situations, such as after mating [101,102]. According to the evaluation of the New Area Test, previously trained animals shook significantly less after handling and the second and third sampling than the animals in the control group. Thus, shaking as a “reordering behavior” in the New Area Test proves that the handling preceding this test as well as the sampling were perceived as less disturbing by the animals of the training versus control group. For the latter, any interaction with humans inflicts a higher disturbance potential than for the animals of the training group due to the lack of conditioning.

#### 4.3.2. Locomotion

A term frequently mentioned in the context of open field tests and especially in the context of chickens, but also rats or mice, is so-called freezing [103,104,105]. This “persistence” is considered a typical behavior of prey animals suddenly confronted with an open, unknown area—an open field—without any protection or cover options. For other animal species, this test should first be verified with regard to the respective species’ biology and species-typical behaviors before it is carried out or evaluated.

Since the New Area Test is often accompanied by the separation of the individual being tested from its group, an increase in the locomotion rate intended to allow the animal to reunite with its group is sometimes shown in contrast to freezing [106,107,108]. However, it has been suggested that this effort to reunite is greater in chicks than in adults and only decreases with age in favor of enemy avoidance tactics such as freezing [109]. Nevertheless, our experiments were expected to primarily observe reunification efforts, since the animals were not tested in a large-scale open field, but in a space adjacent to their home stable with their aviary. This space, referred to here as the New Area, exhibited several similarities with the home stable, such as its spatial and acoustic design. Sounds made by the other group members in the home stable room could be heard by the animal in the New Area Test. As the chicken’s sense of smell is relatively well developed [110,111], the olfactory perception of the other chickens and the stable room itself by the animal in the New Area can also be assumed.

Thus, it was to be expected that the isolated individual would rapidly exert efforts to reunite with its group members in known proximity. This endeavor, for which Vallortigara et al. [112] describe a greater tendency in hens than in roosters in so-called runway tests, should not be mitigated by the presence of a human observing the respective animal in the New Area, which, according to Gallup and Suarez [113] and Gallup et al. [114], may be interpreted as potential enemy, thus representing a trigger for increased freezing. In the present study, this parameter was mitigated by the observer being known to the animals of both groups from the daily feeding and housing routine. The animals in the training group were separated from their group members daily for clicker training. Although each animal currently being trained maintained constant eye contact with the other group members, it was isolated to a certain extent. Nevertheless, this training was not sufficient to strengthen the relaxation of these chickens as a social species to the extent of adaptation. The efforts of the animals of both groups to reunite with their conspecifics, characterized by the movement by animal lengths, were only suggestive of a reduced urge to reunite in the animals of the training group.

#### 4.3.3. New Area Test—Defecation

The correlation between stress and fecal output is evident in the assessment of fecal droppings shed in the New Area Test. The analysis of the number of droppings (Table 6) indicates a tendency toward stress perception, whilst the analysis of the fecal consistency (Table 6) indicates a significantly reduced stress perception in the training versus control group animals. These results closely correspond to those of both the corticosterone measurement and the axial body shaking behavior, and attribute a reduced stress perception to the trained versus untrained animals.

### 4.4. Cannibalism and the Role of Active Cognitive Enrichment

In control groups 1 and 3, the behavioral disorder of cannibalism, known from many chicken farms and sometimes associated with stress [115], appeared in three and two hens, respectively, in the last third of the respective experimental run. Although this work succeeded in standardizing and, if necessary, mitigating the stressors frequently relevant in chicken farms, such as wrong temperature, poor lighting, high stocking density, poor health status, laying period, vaccinations, or feeding regime [35,49,80,81,82,83,84,85,86,87,88,89,90,91,92,93], the cannibalism problem shows that there are not only problems in the health status or housing of the animals that negatively affect their welfare, but rather, it shows how important it is to also allow animals to exercise their native and natural patterns of behavior and activity. Considering the everyday life of a wild chicken in its natural environment, this is characterized above all by constant confrontation with a living and moving environment. The animals in the control groups had the same “environmental enrichment” in their aviary as the animals of the respective training groups, i.e., pecking toys, pecking stones, and suspended feed beets. In this regard, Gvaryahu et al. [116] describe that such enrichment of the housing environment with employment objects can reduce aggressive behavior and deaths in white laying hens. In the choice of employment material for the study conducted here, objects were selected that were primarily adapted to the hens’ species-specific food acquisition behavior. Dealing with the variety of food that the chicken uses, especially in the context of foraging, demands a broad behavioral repertoire and a variety of movement sequences from the chicken every day anew. For the training group animals, this demand was simulated in the study presented here by the clicker training as cognitive enrichment, while the animals of the control groups only engaged with their enrichment objects on their own initiative. Based on this, and in addition to measures that, like clicker training, are intended to mitigate direct negative experimental influences on chickens, future studies must also test measures that not only passively offer the animals the possibility of occupation in their housings, but also those that actively challenge the respective individuals with new stimuli as much and often as possible. Possibly also in connection with artificial intelligence, this type of “active cognitive enrichment” could lead to a more intensive use of the enrichment measures offered and prevent behavioral disorders, thus achieving even more holistic positive effects on the well-being of the animals.

## 5. Conclusions

In considering the results of corticosterone levels with caution, the sum of all assessed parameters shows that a three-week clicker training program may beneficially influence the behavior of chickens kept for experimental purposes. The high motivation and willingness to train of most of the animals in the training group indicate that clicker training, as a supporting measure in animal experiments, can be a suitable method for reducing stress or increasing resilience and, thus, represents an applied refinement in the sense of cognitive enrichment. This is underlined by the fact that cannibalism did not occur in any of the three training groups, whereas it was present in two of the three control groups. In this regard, further enrichment options should also be discussed that do not only offer a measure of passive occupation, e.g., by providing feed balls or pecking stones, but also actively challenge the chickens, which are a very active species by nature. Here, artificial intelligence may also gain importance in the future.

The data collected here warrant confirmation through further studies on the influence of stress perception, which should consider additional parameters to evaluate stress perception, have longer training periods, and should assess situations with higher stress potential. With regard to the use of clicker training as a preparatory or supportive measure for animal experiments, factors such as the degree of stress expected, the duration of the planned experiment, and the training time necessary, which is to be added to the actual experimental period, must be taken into close consideration. Since these factors may lead to unfavorable consequences such as a disproportionate extension of the experimental duration and an increase in or facilitation of other stressors, the principle of proportionality should always underpin the decisional basis.

## Figures and Tables

**Figure 1 animals-13-03836-f001:**
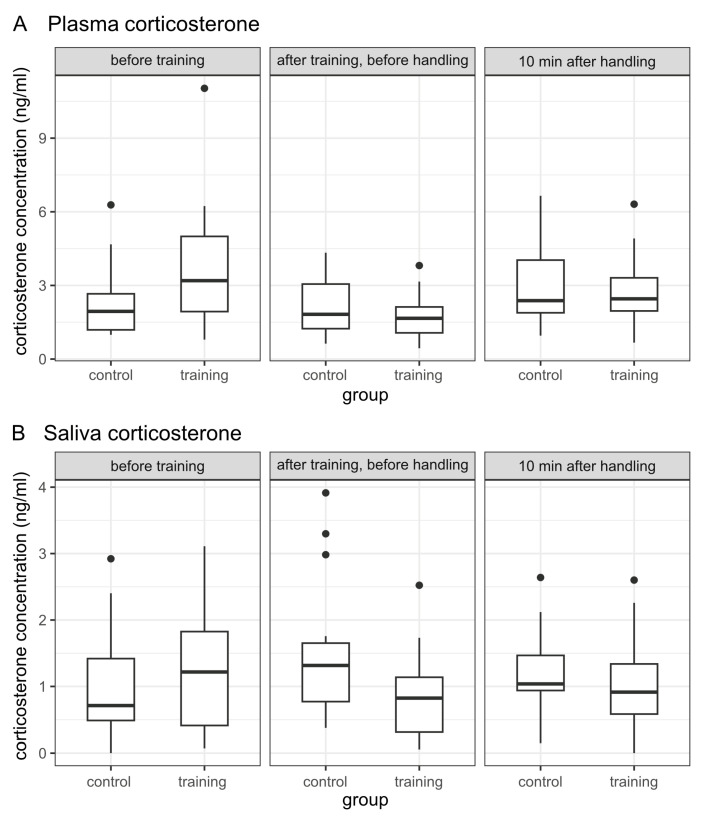
Group comparison based on plasma corticosterone concentration (**A**) and saliva corticosterone concentration (**B**).

**Table 1 animals-13-03836-t001:** Evaluation scheme for the behavior exhibited in the New Area Test.

Behavior Exhibited	Behavior Is Equal
Axial body shaking	→ to an axial shaking movement out and back to the starting position
Locomotion	→ to moving of the chicken by one body length

**Table 2 animals-13-03836-t002:** Fecal parameter assessment.

Fecal Parameters	Type of Assessment
Number of fecal droppings	Ordinal: 0–3
Fecal consistency	Nominal: 1 = well formed, with clearly separated uric acid and fecal components; 2 = pasty, no clear separation of components; 3 = mushy, clearly increased to separate liquid part, pungent odor

**Table 3 animals-13-03836-t003:** Group comparison based on plasma corticosterone concentration.

Plasma Corticosterone								
Sample	Number of Individuals (*n*)		Geometric Mean (ng/mL)		*t*-Test for Independent Samples			
	C	T	C	T	df	*t*-Value	*p*-Value	Cohen’s d
1	21	21	1.998	2.948	40	2.106	0.042	0.650
2	21	21	1.832	1.524	40	−1.057	0.297	−0.326
3	21	21	2.581	2.450	40	−0.323	0.748	−0.100

C = control group; T = training group; sample 1: taken prior to three-week training; sample 2: taken after three-week training, but prior to 10 min handling; sample 3: taken after 10 min handling.

**Table 4 animals-13-03836-t004:** Group comparison based on the saliva corticosterone concentration.

Saliva Corticosterone								
Sample	Number of Individuals (*n*)		Median (ng/mL)		Mann–Whitney U Test			
	C	T	C	T	U	Z	*p*-Value	Effect Size
1	20	20	0.713	1.218	167.5	−0.879	0.379	0.139
2	21	20	1.316	0.825	123.0	−2.269	0.023	0.354
3	21	20	1.038	0.915	169.0	−1.069	0.285	0.167

C = control group; T = training group; sample 1: taken prior to three-week training; sample 2: taken after three-week training, but prior to 10 min handling; sample 3: taken after 10 min handling.

**Table 5 animals-13-03836-t005:** Comparison of the two groups based on their behavior in the New Area Test.

**Axial Body Shaking**							
**Number of Individuals * (*n*)**		**Median**		**Mann–Whitney U Test**			
**C**	**T**	**C**	**T**	**U**	**Z**	***p*-Value**	**Effect Size**
20	17	2.5	1.5	91.5	−2.413	0.016	0.400
**Locomotion**							
**Number of Individuals * (*n*)**		**Mean**		***t*-Test for** **Independent Samples**			
**C**	**T**	**C**	**T**	**df**	**t-Value**	***p*-Value**	**Cohen’s d**
20	17	20.05	18.18	35	−0.938	0.345	−0.310

C = control group; T = training group. * The different numbers of individuals per group are due to the following reasons:

**Table 6 animals-13-03836-t006:** Defecation in the New Area Test.

Defecation in the New Area Test								
Parameter	Number of Individuals (*n*)		Median		Mann–Whitney U Test			
	C	T	C	T	U	Z	*p*-Value	Effect Size
**Number of droppings deposited (0–3)**	21	21	2	2	170.5	−1.378	0.168	0.213
**Consistency of droppings (Scoring: 1–3)**	21	20	1.67	1	54.0	−4.584	0.000	0.716

C = control group; T = training group.

## Data Availability

The data presented in this study are available on request from the corresponding author. The data are not publicly available due to further planned analyses.

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
