# Peer review of "Clicker Training as an Applied Refinement Measure in Chickens"

_animals, 2023, doi:10.3390/ani13243836_

Round 1
Reviewer 1 Report
Comments and Suggestions for Authors
Dear Author(s):
- Abstract and keywords are sufficient and include the study.
- The introduction is very long. Too many references are used. Must be rearranged.
- If there is a photo for Table 1 and Table 2, it can be included in the article.
- The method is written in a clear and understandable way. This situation is enough.
- It would be more appropriate to write the Results section in the same order as the method section.
- Discussion and conclusion are sufficient. Includes study findings. It is beautifully written.
- It would be good to make the necessary arrangements so that the work will have a better effect and readability.
Author Response
Please see in the attached file.
Reviewer 2 Report
Comments and Suggestions for Authors
This paper represents in general an interesting study, however some things needs to be clarified and backed-up by literature. Major points are the usage of Cort measure that were below detection threshold and that the results from the clicker training is missing. All comments are laid out in detail below and may be solved in a revision.
- Lines 39-41: Please start with a full sentence and maybe merge these lines.
- Lines 58-59: You shall here also mention an intensive operant conditioning study on chicken, i.e. Dudde et al., 2018 Frontiers in Psychology, 9, 2000.
- Line 59: It must not always be directly a trainer but can also be an automated routine, see reference mentioned above.
- Lines 59-64: The clicker sound is only one potential stimulus for the operant condition, it is worth to mention also others
- Lines 66-69: As the manuscript is aiming to an international audience, here at least it should be referred to broader number of the EU
- Line 90: As you refer above more broader to birds I would suggest to include also other than only chicken here. A study from zebra finches and CORT as a stress measure might be e.g. Krause & Ruploh - Applied Animal Behaviour Science, 182, 80-85 or Honarmand et al., 2010 PLoS One, 5(9), e12930.
- Lines 105-106: Another important test is also the Tonic Immobility test Gallup, G. G. (1979). Animal Behaviour 27(1), 316–317, which should at least briefly be mentioned, and probably one or two sentences are worth to include why the open field test might be better suited here
- Line 132: I really appreciate the combination of serological and ethological parameters, it really strengthens the data
- Lines 155-166: Please include here the number of birds tested in the study, and if different how many birds were in the flock(s) in total. And were all birds kept in one group. How much space was provided per birds? And please specify a bit the rudimentary nutritional program. At what age did testing start?
- Lines 177-178: Have this kind of reward been used in studies before, if so, please refer here to them.
- Lines 186-187: Please provide reference for this statement.
- Line 196: Was the trainer always the same person?
- Lines 241-242: How long was collection time at min and at max for each bird per group?
- Table 1: Locomotion definition? Was the one body length taken as parameter regardless where the motion took place or distance to something / someone?; And omit “Behaviour of reordering” as term, just label it axial shaking
- Lines 319-322: I agree with the tests used, but could you please indicate for which parameters and variables tests were calculated?
- Lines 336: When samples were below detection threshold the minimum concentration was considered? If so, this is not appropriate here as you do not know whether concentration is indeed so low or whether something in sample handling / analysis has gone wrong. These data points must be omitted, you can not invent numbers. For how many samples was this the case? Please exclude these data points and recalculate the tests.
- Lines 332-355: This should entirely moved to the statistical method section, rather than being located in the results
- Lines 356-368: Description of the findings is required along the Tables and Figures, please add. Furthermore, please add to Table 3 and 4 , also the statistical values like Z and t- values and degrees of freedom
- Figure 1: Please add a line for the y-axis and omit colours and better split for the three samples into a,b,c panels. Also omit please the vertical lines in the graph and adjust scaling labels of y-axis
- Figure 2: Omit please vertical lines and colours of bars and split in three panels as suggested for Figure 1
- - Figure 1 and 2 can be omitted in total as data is fully shown in the Tables, no need to double illustrate date
- Lines 371-378: Move to statistical methods and replace by a description of results that are apparent in Table 5 and Figures 3,4.
- Table 5 add t and Z -values and degrees of freedom for the stats
- Figure 3 and 4: Omit vertical lines please
- Figure 3 and 4 can be omitted in total as data is fully shown in the Tables, no need to double illustrate date
- Lines 397-403: Move this to statistical methods and add here a description of the results shown in Table 6 and Figures 5,6
- Figure 5 and 6: Omit vertical lines please, where is the median in Fig. 6 for the training group?
- Figure 5 and 6 can be omitted in total as data is fully shown in the Tables, no need to double illustrate date
- Table 6 add Z -values and degrees of freedom for the stats
- Lines 413: What is the positive effect on stress response, none in the plasma Cort and only in sample 2 in the saliva cort? Thus, wouldn’t agree to this clear statement here.
- Line 415: Was supports you that it is a cognitive enrichment? I have seen no data on success of the clicker training, do the birds learn it at all to make the association between clicker and reward? Please add these results here.
- Line 451: As requested above, please indicate min and max times for sampling duration.
- Lines 446-449: I do not get what you aim to say here.
- Lines 457-459: No, there was no such trend, not statistically and even not on mean level. You could not make this statement.
- Lines 460-470: Only one of the four tests in the expected direction, thus statement might be formulated much more cautions and alternative explantions, i.e. false positive finding in one case must at least be mentioned and briefly discussed. I agree that it is not likely but at least be discussed.
- Lines 473-475: Would not agree to this as in the other case the skin is punctured, do you have any reference to this.
- Lines 509-512: Do you have any reference for this interpretation?
- Line 600: With respect to cognitive enrichment, this is not really well discussed and see my comment to line 415
Author Response
Please see in the attached file.

Reviewer 3 Report
Comments and Suggestions for Authors
The Document is well written and covers an important and novel topic of training chicken in experimental surroundings - in preparation for animal testing. The aim is to reduce stress in animal experimentation, therefore this paper very important for the field of animal welfare. Generally, the training protocol is small-stepped and reasonable. I also like the additional information about cannibalism –considering cognitive enrichment in animal experimentation and medical training.
This is a good alternative research approach and sounds promising.
Please use either “body shaking” or “body shacking” throughout the manuscript and be consistent in terminology.
Material and Methods:
- More details on housing, barn or aviary system? In Line 159 you mention a barn system and later on an aviary. Please describe the housing system in more detail, were they housed in groups? How many hens in one group?
Comments in Detail:
35-36 clicker training, positive reinforcement training
60 The animal can not only show it more often but it can also be taught to perform it on a signal from the trainer
105-114 This test only works because chickens and rats are prey animals and their evolutionary nature makes them afraid of open spaces. So the test works because they are confronted with a fear inducing situation. Maybe explain better that the test is supposed to simulate a stressful situation? It should also be mentioned that this test can therefore not be readily transferred to other species, but its function must first be verified.
143-144 How can you know why the animal moves in the test and if not moving is positive or negative for the animal?
155-172 Mention acclimation period?
169-172 How long did one training session take? For me the information is lacking that the clicker training is supposed to train exactly the behaviors needed for in the “stress situation” in order to prepare them for that and is not just a random training of fun behaviors to just make them used to people. The training is already a kind of medical training where you gradually train unpleasant procedures
182-183 I don’t understand how the gadget looks like in the end, the cup glued on the outside of the spoon? Maybe a picture would be useful?
185 In the paragraph below not only influence of the breed is explained but also influence of the individual, you might want to change the subtitle
197, 199, 202, 205 Simultaneously or directly after?
207 What exactly is the “area of the chickens”? Were the chickens eating while being touched? Which body parts were touched?
222 “Regardless of where the trainer positioned it and himself in the room”… ? did the trainer still had to stand behind the table or could he stay at the other end of the room?
226 Type -O, please change Weighting to “weighing”?
225-236 The group training was described very detailed, but this is described only shortly and superficially…
241 How many people were doing sampling collection and training?
244 What exactly happened in the stress situation and in which order?
252 One or more tubes of blood per animal? Was the time of the day considered in sampling for corticosterone?
298 How did the corridor look like? What size? Could they see, hear or smell their group mates? Was there natural or artificial light?
299-300 Each video sequence was coded by 2 persons? Was the inter-observer reliability tested?
300-304 Why exactly did you decide to evaluate only these behavioral parameters and not e.g. vocalization as well?
305 In both New Area Tests?? When is the second one done?
Results
344-355 What about the results of samples 1? It seems that Control animals have lower basic Corticosterone concentrations esp. in plasma but also slightly in saliva then the training group prior to training. This should be put into relation with the results of sample 2 and 3. The concentrations of the trained animals lowered much more than the ones of the control animals rose. Was a statistical test used that takes the repeated testing of the animals into account?
429 ff to which extend did you consider the effects of daytime and lighting in your results?
430 Are you sure that the source 77 you mention is also on stocking density or ammonium concentration?
496-502 One sampling team did the 7 animals of one group one after the other and at the same time the second team in the other group. This does safe time for the humans, but not for the animal groups. Wouldn’t it be more effective to safe time for the animals if the teams worked on the same animal group at the same time?
508 ff in other mammalian species body shaking is know to be a calming signal (eg in dogs). Could this be transferred to chicken as well?
531 Tested = separated?
541-547 Why did the person stay in the room if the behavior was analyzed by video recording anyways?
463 Headline 4.1 – 4.3 is underlined, this one not
727 Meat Qu ality; please remove the space
730 variouspre-sampling; there is a space missing
Figure 1 and Figure 2 present the same information as Table 3 and 4. Please remove either one. Same for Figure 3 and 4 or Table 5 and 6; Figure 5 and 6 and Table 6. Please either present the information in Tables OR Figures.
Author Response
Please see in the attached file.
